# Cytoplasmic dynein crosslinks and slides anti-parallel microtubules using its two motor domains

**Marvin E Tanenbaum[†], Ronald D Vale*, Richard J McKenney[†]**

Department Cellular and Molecular Pharmacology, Howard Hughes Medical Institute, University of California, San Francisco, San Francisco, United States

**Abstract** Cytoplasmic dynein is the predominant minus-end-directed microtubule (MT) motor in most eukaryotic cells. In addition to transporting vesicular cargos, dynein helps to organize MTs within MT networks such as mitotic spindles. How dynein performs such non-canonical functions is unknown. Here we demonstrate that dynein crosslinks and slides anti-parallel MTs in vitro. Surprisingly, a minimal dimeric motor lacking a tail domain and associated subunits can cause MT sliding. Single molecule imaging reveals that motors pause and frequently reverse direction when encountering an anti-parallel MT overlap, suggesting that the two motor domains can bind both MTs simultaneously. In the mitotic spindle, inward microtubule sliding by dynein counteracts outward sliding generated by kinesin-5, and we show that a tailless, dimeric motor is sufficient to drive this activity in mammalian cells. Our results identify an unexpected mechanism for dynein-driven microtubule sliding, which differs from filament sliding mechanisms described for other motor proteins.

**\*For correspondence:** vale@cmp.ucsf.edu

[†]These authors contributed equally to this work

**Competing interests:** The authors declare that no competing interests exist.

**Reviewing editor**: Tony Hyman, Max Planck Institute of Molecular Cell Biology and Genetics, Germany

## Introduction

Cytoplasmic dynein is a 1.2 MDa, multisubunit microtubule motor complex that belongs to the AAA family of molecular machines (*Ogura and Wilkinson, 2001*; *Allan, 2011*). The dynein complex is composed of a dimer of two heavy chains. Each heavy chain contains a C-terminal motor domain, and an N-terminal tail domain that binds to accessory chains and adaptor proteins, which are needed to link the motor to its cargo (*Kardon and Vale, 2009*). Dynein transports a plethora of cargoes towards MT minus-ends, including many types of organelles, mRNAs and proteins.

In addition to its well-studied role in cargo transport, cytoplasmic dynein has been implicated in the organization of the MT cytoskeleton itself, particularly during cell division. When mammalian cells enter mitosis, dynein is needed to remodel the prophase MT network (*Rusan et al., 2002*), and at later stages dynein assists in organizing MTs to form focused spindle poles (*Heald et al., 1996*; *Merdes et al., 1996*). Dynein also generates an inward force within the spindle that counteracts an outward force generated by kinesin-5 and kinesin-12 motors (*Mitchison et al., 2005*; *Tanenbaum et al., 2008*, *2009*; *Ferenz et al., 2009*; *Vanneste et al., 2009*). The balance of these forces is important for normal spindle assembly (*Tanenbaum and Medema, 2010*). In cell extracts, dynein also has been shown to organize MTs into aster-like structures (*Verde et al., 1991*; *Gaglio et al., 1996*), drive the fusion of two closely positioned spindles into a single bipolar spindle (*Gatlin et al., 2009*), and transport stabilized MT seeds to the spindle pole (*Heald et al., 1996*). Collectively, these results show that dynein plays an important role in organizing the MT network during cell division.

In contrast to cargo transport, which involves a well-studied walking mechanism of the two dynein motor domains along a MT (*Gennerich and Vale, 2009*; *DeWitt et al., 2012*; *Qiu et al., 2012*), the mechanism by which dynein organizes MTs in a MT network has not been established. One possibility is that dynein is anchored at fixed subcellular sites through its tail domain and moves processively along MTs through the cytoplasm. However, since the 'cargo' is sufficiently large in this instance,

**eLife digest** When cells divide, they must also divide their contents. In particular, both 'mother' and 'daughter' cells require full sets of chromosomes, which must first be duplicated, and then evenly distributed between the cells. Protein filaments called microtubules form a network that helps to accurately segregate the chromosomes. Microtubules emanate from structures at each end of the dividing cell known as spindle poles; after the chromosomes have duplicated, the microtubules latch onto them and align the pairs in the middle of the cell. As the two cells separate, microtubules at opposite spindle poles reel in one chromosome from each pair.

Microtubules are composed of alternating copies of two different types of a protein called tubulin, and have ends with distinct properties. The 'minus' ends are directed outwards, away from the chromosomes; the 'plus' ends—which can actively add tubulin—grow toward the middle of the cell, and can also bind to chromosomes. Microtubules can be manipulated by motor proteins that 'walk' along them carrying cargoes, which can include other microtubules. The combined actions of many motor proteins rearrange the microtubule network into a configuration that enables the chromosomes, and other cellular structures, to partition equally between the mother and daughter cells.

Motor proteins such as dynein and kinesin transport cargoes along microtubules; each motor is composed of two identical copies of the protein bound to each other. Kinesin walks toward the plus end of a microtubule, propelling itself using 'feet' that are called motor domains; it binds cargoes (including other microtubules) through additional regions located at the opposite end of the protein. In contrast, dynein walks toward the minus end of a microtubule. Although dynein is known to carry certain cargoes through regions outside its motor domain, how it transports other microtubules is not well understood.

Tanenbaum et al. now show that regions outside the motor domain of dynein are unnecessary to transport microtubule cargoes. When two dynein motor domains are isolated and linked to each other in vitro, each can bind to a separate microtubule. By walking toward the minus ends of their respective microtubules, the motor domains drive the microtubules in opposite directions, sliding them apart. These studies thus provide insight into the mechanism through which dynein works with additional motor proteins (such as kinesin) to rearrange microtubules during cell division—and also to ensure that chromosomes segregate evenly between mother and daughter cells.

the MT themselves would move, rather than the cargo (*Wühr et al., 2009*). Such a mechanism may also function during spindle positioning, where cortically-anchored dynein pulls on spindle MTs (*Galli and van den Heuvel, 2008*; *Kiyomitsu and Cheeseman, 2012*; *Kotak et al., 2012*; *Laan et al., 2012*), as well as during centrosome separation, where dynein is anchored to the nuclear envelope (*Splinter et al., 2010*; *Bolhy et al., 2011*; *Raaijmakers et al., 2012*).

Alternatively, dynein could physically crosslink two MTs and slide them relative to each other. To generate sliding between two MTs, an individual dynein motor, a dimer of two heavy chains, could use its two motor domains to walk along one MT and employ a second MT binding domain to transport a 'cargo MT'. Axonemal dyneins function in this manner to produce MT sliding in cilia and flagella. Several classes of kinesin utilize a similar mechanism for crosslinking and sliding MTs (*Straube et al., 2006*; *Braun et al., 2009*; *Fink et al., 2009*; *Seeger and Rice, 2010*; *Su et al., 2011*; *Weaver et al., 2011*). However, there is currently no evidence suggesting that cytoplasmic dynein contains a secondary MT binding site outside of the motor domain. A second model is that dynein could crossbridge and slide MTs by forming small oligomers or co-complexes with other proteins. Kinesin-5, which forms a bipolar tetramer of motor domains, crossbridges and slides MTs by such a mechanism (*Kashina et al., 1996*; *Kapitein et al., 2005*). A third model is that dynein's two motor domains bind to and move along two separate MTs.

Here using an in vitro assay, we demonstrate that both native rat, and recombinant yeast cytoplasmic dynein can drive sliding of anti-parallel MTs in the absence of additional proteins. Single molecule data of dynein in a MT overlap zone is most consistent with a model in which the two dynein motor domains walk along different MTs to generate sliding. We also show in vivo that a minimal dynein dimer, lacking expected cargo binding interactions, can substitute for native dynein in generating an inward force

within the spindle that antagonizes outward MT sliding forces generated by kinesin-5 and kinesin-12. Together, these results show that dynein can slide and organize MTs, using a sliding mechanism that differs from that described for other motor proteins.

## Results

### Dynein crosslinks and slides MTs within bundles

Previous work has reported that purified brain cytoplasmic dynein crosslinks and bundles MTs (*Amos, 1989*; *Toba and Toyoshima, 2004*), but the mechanism has remained unclear. We similarly found that purified rat brain dynein induces the formation of large bundles of purified MTs (*Figure 1A–D*). Since cytoplasmic dynein can crossbridge MTs, we next investigated whether it can slide two MTs relative to each other. To test for MT sliding, we incubated brain dynein in solution with both green- and red-labeled fluorescent MTs; the green MTs in the MT bundles were also biotinylated, allowing their immobilization onto a streptavidin-coated coverslip (*Figure 1D,E*). Addition of ATP into the assay chamber induced the red MTs to slide within the bundles relative to the stationary green biotin-MTs (*Figure 1E*; *Video 1*).

To crosslink and slide MTs, dynein could bind one MT with its motor domains and potentially another with a non-motor MT-binding domain (*Figure 1D*). Alternatively, the two motor domains of the dynein dimer could each bind a different MT (*Figure 1D*). To distinguish between these possibilities, we tested a well characterized, GST-dimerized, truncated yeast dynein construct, GST-Dyn1$_{331kDa}$, which contains the motor domain but lacks the non-motor tail domain and other dynein subunits (*Reck-Peterson et al., 2006*) (*Figure 1A,B*). This minimal, dimeric dynein was affinity purified, followed by gel filtration to remove any potential oligomers or aggregates. GST-Dyn1$_{331kDa}$ crosslinked MTs, while a monomeric version lacking the GST dimerizing domain did not (*Figure 1A–C*). Similar to rat brain dynein, the GST-Dyn1$_{331kDa}$ motor induced sliding of MTs within the bundles, when ATP was added (*Figure 1F*; *Video 2*). These results provide the first direct in vitro demonstration that cytoplasmic dynein can slide MTs within bundles. They also reveal that sliding does not require dynein associated subunits or any potential second MT binding domain in the dynein heavy chain tail domain.

### Cytoplasmic dynein slides two anti-parallel MTs

We next sought to develop an improved MT sliding assay in which the relative movement of two MTs could be more easily observed. To achieve this, we first bound GST-Dyn1$_{331kDa}$ in the absence of ATP to an immobilized 'track' MT on the coverslip surface, and then introduced 'transport' MTs, which became crosslinked to the 'track' MT by dynein. After addition of ATP, GST-Dyn1$_{331kDa}$ drove robust sliding of transport MTs along the immobilized track MTs (*Figure 2A*, *Video 3*). The transport MTs frequently moved until they reached the end of the track MT, where they swiveled around an end point on the track MT (asterisk *Figure 2A*). The speed of MT-MT sliding (53 ± 24 nm/s; mean ± SD; *Figure 2A*) was similar to the velocity observed in MT gliding assays (47 ± 11 nm/s; *Figure 2A*) with this dynein construct. We found that purified rat brain dynein also produced MT-MT sliding using this same assay, although the movement was slower than surface gliding (10 ± 5 nm/s vs 615 ± 200 nm/s) and characterized by more frequent pausing than MT gliding by surface-bound brain dynein (*Figure 2B*, *Video 4*). The slower motility may be a result of the much weaker processivity and directionality observed for individual mammalian dyneins in vitro compared to yeast dynein (*Ross et al., 2006*; *Cho et al., 2008*; *Miura et al., 2010*; *Ori-McKenney et al., 2010*; *Trokter et al., 2012*). Other proteins (such as LIS1 and Ndel1) that directly modify these properties of mammalian dynein may be needed for more robust processivity and sliding, consistent with the requirement of these proteins for dynein-dependent MT organization in vivo (*Raaijmakers et al., 2013*). Nonetheless, the finding that rat dynein alone is capable of crosslinking and sliding MTs in our assays, shows that this type of motility is an intrinsic ability of the motor.

To determine the orientation of MT-MT sliding driven by dynein, we repeated the assay described above using polarity-marked MTs and GST-Dyn1$_{331kDa}$. These experiments revealed that the large majority of MT sliding events involved MTs in an anti-parallel configuration (18 of 21 events, *Figure 2C*). To confirm this, we also analyzed MT polarity using the direction of movement of single GFP-tagged dynein motors on the track and transport MT, in cases where the transport MT moved past the end of the track MT. In 21 of 22 cases, the transport and track MTs were anti-parallel to one another. This result is consistent with a model in which dynein binds to an anti-parallel MT overlap with one motor

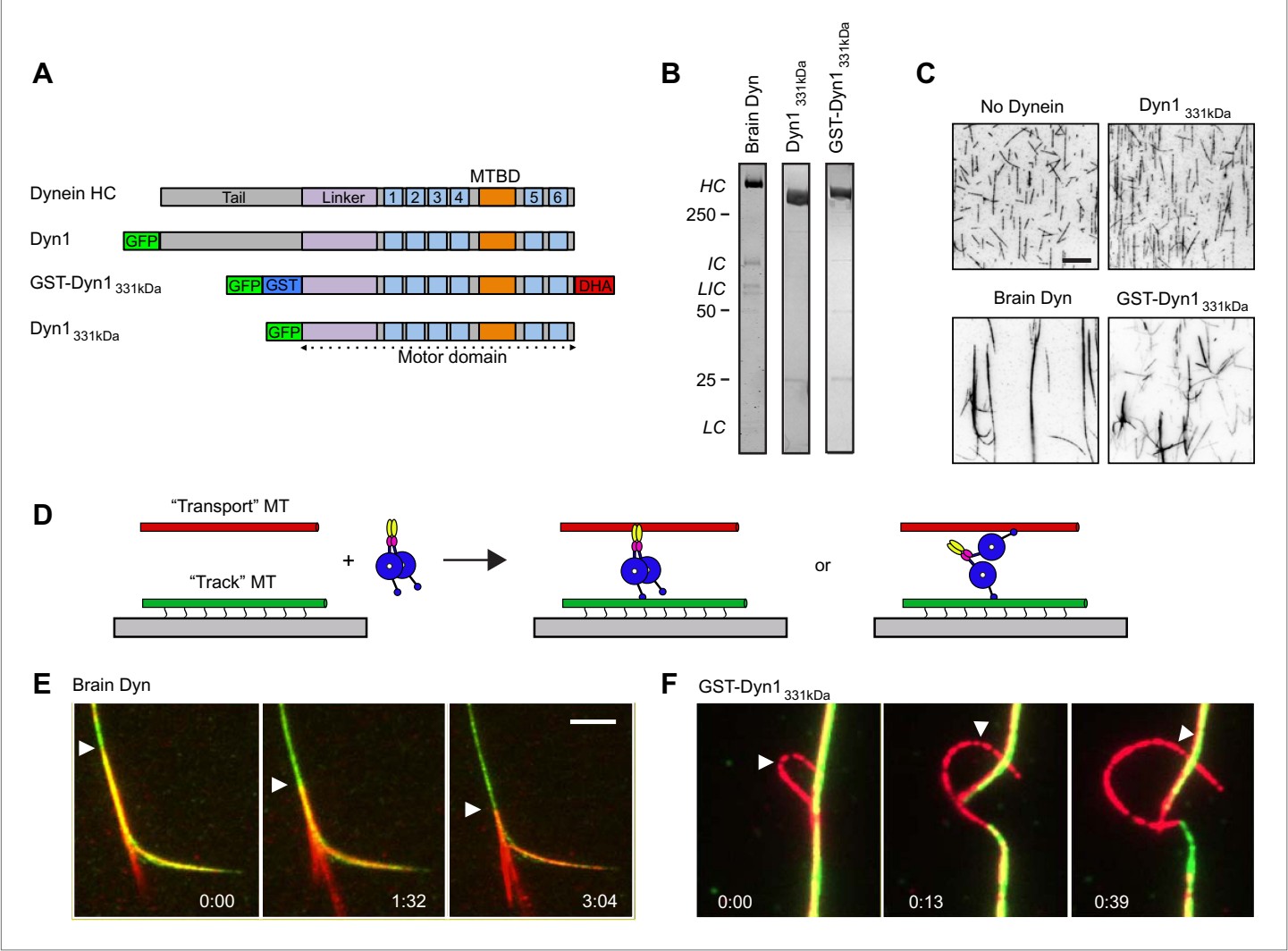

**Figure 1**. Dynein crosslinks and slides MTs in bundles. (**A**) Schematic overview of the dynein constructs used in this study. The N-terminal tail is shown in gray, the linker in purple, the six numbered AAA+ domains are in light blue and the stalk and MT binding domain are depicted in orange. GFP and GST tags are shown in green and blue, respectively. The Halo tag (DHA, Promega) is shown in red. (**B**) Coomassie brilliant blue stained gels showing purified dynein constructs used in this study. The associated subunits of the brain cytoplasmic dynein complex are labeled; HC–heavy chain, IC–intermediate chain, LIC–light intermediate chain, LC–light chain. Recombinant yeast dynein constructs do not contain associated subunits. Molecular weight markers are indicated. (**C**) MTs incubated in the absence or presence of dynein are visualized by attachment to a streptavidin-coated coverslip via a biotin tag. Brain dynein and GST-Dyn1$_{331kDa}$ crosslink MTs into large bundles, while the dynein monomer, Dyn1$_{331kDa}$ does not. Scale bar, 10 μm. (**D**) Cartoon depicting two different mechanisms by which dynein could crosslink MTs, either using its two motor domains or through the tail domain. Alexa-568 and Alexa-488 labeled MTs are crosslinked by dynein. The green MTs are attached to the coverslip through a biotin-streptavidin linkage and perfusion of 1 mM ATP induces sliding between the MTs. (**E** and **F**) Example of rat (**E**) and GST-Dyn1$_{331kDa}$ (**F**) dynein-driven sliding of red-labeled MTs within the bundle after 1 mM ATP addition. Arrowhead tracks the sliding MT within the bundle. The time relative to the start is noted in min:s at the bottom of each image. Scale bar, 5 μm.

domain on each MT in the overlap. In this configuration, both motor domains walk towards the minus-end of their respective MT, thereby driving sliding of the crosslinked pair of MTs (**Figure 1D**).

## A natively dimerized, minimal dynein crosslinks, and slides MTs

The above results demonstrate that GST-dimerized yeast dynein, and the native rat dynein complex, both can crosslink and slide MTs. However, in the case of the minimal, GST-dimerized dynein (**Cho and Vale, 2012**), it is possible that the artificial, anti-parallel dimerization may allow the two motor domains to crosslink MTs in a manner that would not occur in native dynein dimer. Additionally, the native dynein dimer may utilize a different mechanism, in which it crossbridges MTs through an additional MT

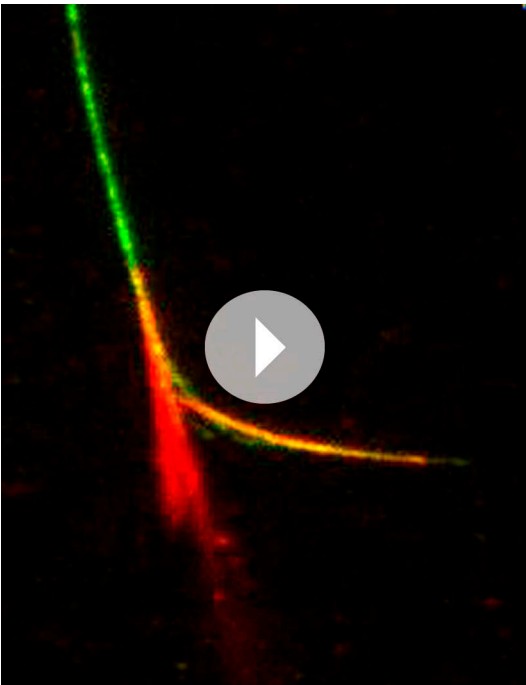

**Video 1**. Rat brain cytoplasmic dynein bundles and slides MTs. Red-labeled MTs are slid and extruded from a bundle of red- and green-labled MT bundles by rat brain cytoplasmic dynein upon addition of ATP. Total time of the video is 551 s. Playback is 30 fps.

binding site, either in the tail of the heavy chain or in one of the dynein accessory subunits. To rigorously examine these possibilities, we tested a truncated dynein molecule that includes its native dimerization domain, but lacks any of its associated subunits. GST-Dyn1$_{331kDa}$, is monomeric in the absence of GST (*Reck-Peterson et al., 2006*), so we reasoned that a larger portion of the tail domain was necessary for dynein heavy chain dimerization. Prior unpublished observations from our lab indicated that a dynein with a slightly longer N-terminus (Dyn1$_{387kD}$, *Figure 3A*) was processive when fused to GFP (S Reck-Peterson and A Carter, unpublished observations), and processivity is an attribute of a dynein dimer (*Reck-Peterson et al., 2006*). We overexpressed Dyn1$_{387kD}$ from a Gal promoter, and the purified protein revealed no co-purifying accessory subunits by silver staining after SDS-PAGE (*Figure 3B*). In sucrose gradients, purified Dyn1$_{387kD}$ sedimented at approximately 19S (*Figure 3C*), which is similar in size to the full dynein complex (*Paschal et al., 1987*), as well as a truncated dimeric dynein (*Trokter et al., 2012*). Consistent with it being a dimer, single molecule assays revealed robust processive motion of Dyn1$_{387kD}$ along microtubules with an average speed of 75 ± 40 nm/s (n = 95) (*Figure 3D*). Analysis of moving Dyn1$_{387kD}$ molecules revealed a very similar single molecule intensity distribution to the entire Dyn1$_{387kD}$ population (*Figure 3E*), indicating that the moving particles were not a minor subset of aggregated or oligomerized molecules. Further excluding the presence of multimers, the fluorescence intensities of single gel-filtered Dyn1$_{387kD}$ molecules were comparable or even lower in brightness to GST-Dyn1$_{331kD}$ (*Figure 3F*). The reason for the lower brightness of GFP in Dyn1$_{387kD}$ is not clear, although it might be due to fluorescence quenching of nearby fluorophores. Regardless, this result argues that the molecules are not aggregated, and movement is due to dimers and not higher order molecular entities.

Dyn1$_{387kD}$ bundled MTs in the absence of ATP (*Figure 3—figure supplement 1A*), demonstrating that it can crosslink MTs, similar to GST-Dyn1$_{331kD}$. Addition of ATP to MT bundles resulted in MT sliding of MTs in the bundle (*Figure 3—figure supplement 1B*; *Video 5*). Similarly, Dyn1$_{387kD}$ drove efficient MT-MT sliding in single MT overlaps, at comparable speeds to its surface gliding motility (*Figure 3G*; *Video 6*). Thus, Dyn1$_{387kD}$ behaves very similarly to GST-Dyn1$_{331kD}$. Further, the fluorescent dynein molecules appeared homogenous during the sliding event (*Video 7*), ruling out the presence of aggregates within the sliding overlaps.

Finally, we wished to determine if Dyn1$_{387kD}$ might cause MT-MT sliding by employing a second type of MT binding site outside of the canonical MT binding domain (MTBD) in its motor domain (*Figure 3A*) (*Carter et al., 2008*; *Redwine et al., 2012*). To test this possibility, we deleted the canonical MTBD and expressed the protein (Dyn1$_{387kD \Delta MTBD}$). Dyn1$_{387kD \Delta MTBD}$ showed no specific association with MTs in a cosedimentation assay (*Figure 3—figure supplement 1C,D*) and was unable to crosslink MTs (*Figure 3—figure supplement 1E*). Thus, Dyn1$_{387kD \Delta MTBD}$ does not possess a MT binding region outside of the motor domain. Together, these results show that a natively dimerized dynein can crosslink and slide MTs in vitro, and that associated chains or an additional MT binding site is not required for this activity. These results strongly support a model in which dynein slides MTs in vitro using only its two motor domains (*Figure 1D*).

## Dimeric dynein molecules can bind two MTs simultaneously

The previous experiments suggest that a dimer of two motor domains can bind two different MTs simultaneously and generate anti-parallel sliding. This model would predict that individual dimeric

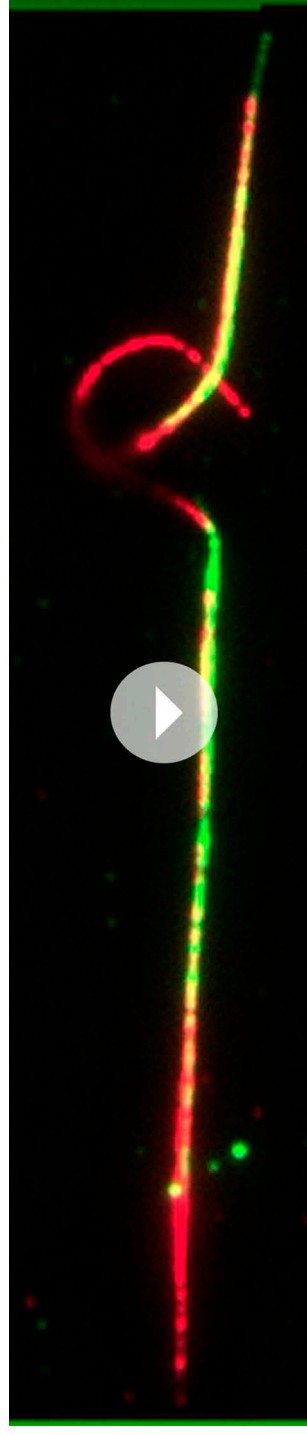

**Video 2**. GST-Dyn1$_{331kDa}$ slides MTs within bundles. Red-labeled MTs are extruded from a bundle of red- and green-labeled MTs by GST-Dyn1$_{331kDa}$ upon addition of ATP, indicating the dynein motor domains alone are sufficient for this activity. Total time of the video is 84 s. Playback is 5 fps.

dynein motors would be approximately stationary in a MT overlap zone, as each motor domain generates opposing pulling forces on the two MTs. To test this model, we imaged individual, fluorescently-labeled dynein molecules within an anti-parallel MT overlap. To generate stable anti-parallel overlaps, we crosslinked MTs with GFP-Ase1 (*Janson et al., 2007*), a protein that crosslinks MTs into anti-parallel bundles with an inter-MT spacing of around 15–40 nm (*Gaillard et al., 2008*; *Roque et al., 2010*; *Subramanian et al., 2010*). This inter-MT distance is within the size range expected for MT crosslinking by a dimeric dynein motor (*Vallee et al., 1988*; *Burgess et al., 2003*). We found that GFP-Ase1 specifically localized to regions of MT overlap (data not shown), as previously described (*Loïodice et al., 2005*; *Yamashita et al., 2005*; *Janson et al., 2007*). Quantitative imaging of MT fluorescence revealed that 2–3 MTs were typically present in each bundle.

As expected from prior work (*Reck-Peterson et al., 2006*), individual TMR-labeled GST-Dyn1$_{331kDa}$ moved unidirectionally and highly processively on single MTs. However, GST-Dyn1$_{331kDa}$ appeared largely immobile in MT overlaps (*Figure 4A*). Interestingly, when individual molecules moving processively along a single MT reached the MT overlap zone, they would appear to stop moving (*Figure 4B*). However, when individual molecules were tracked with ~10 nm precision, it was apparent that individual dynein molecules were not completely immobile in MT overlap zones, but rather frequently moved bidirectionally with long pauses between each run (*Figure 4C*). While yeast dynein occasionally takes backwards steps (*Reck-Peterson et al., 2006*), such long distance directional switching and frequent prolonged pausing were never observed when dynein moved along single MTs (*Figure 4C*).

To rule out that the unusual motility of dynein in MT overlaps was due to the presence of GFP-Ase1 on MTs, we loaded single, surface-immobilized MTs with 10- to 20-fold higher concentrations of GFP-Ase1 than was present in MT overlaps. Despite the high concentration of GFP-Ase1, single dynein molecules moved unidirectionally and processively (*Figure 4D*). Thus, we conclude that bidirectional switching and pausing are due to the presence of a MT overlap, not to the presence of GFP-Ase1 on MTs. To exclude the possibility that the artificial GST-dimerization of the dynein motor domains was affecting dynein motility within the MT overlaps, we repeated these experiments using GFP-tagged full-length yeast dynein (Dyn1) (*Gennerich et al., 2007*) and Dyn1$_{387kD}$. Dyn1 showed processive, unidirectional motility along single MTs (*Figure 4E*), but transitioned to frequent bidirectional switching and pausing within MT overlaps (*Figure 4E*). Similar results were found for Dyn1$_{387kD}$ (*Figure 4—figure supplement 1*).

Taken together, these results strongly suggest that the dynein motor domains have sufficient flexibility to allow two distinct modes of stepping. The two motor domains of the dimer can bind to the same MT track, most likely in a compact, side-by-side configuration of the two AAA ring domains (*Cho and Vale, 2012*), and exhibit persistent unidirectional motion. Alternatively, they can bind to neighboring MTs within a MT overlap and produce a force that acts to slide the two MTs relative to each other (*Figure 1D*).

## A minimal dynein dimer induces spindle collapse in vivo

We next set out to test whether dynein can slide anti-parallel MTs in vivo using only its motor domains. In human cells, kinesin-5, and kinesin-12 motors promote spindle bipolarity by generating an outward force

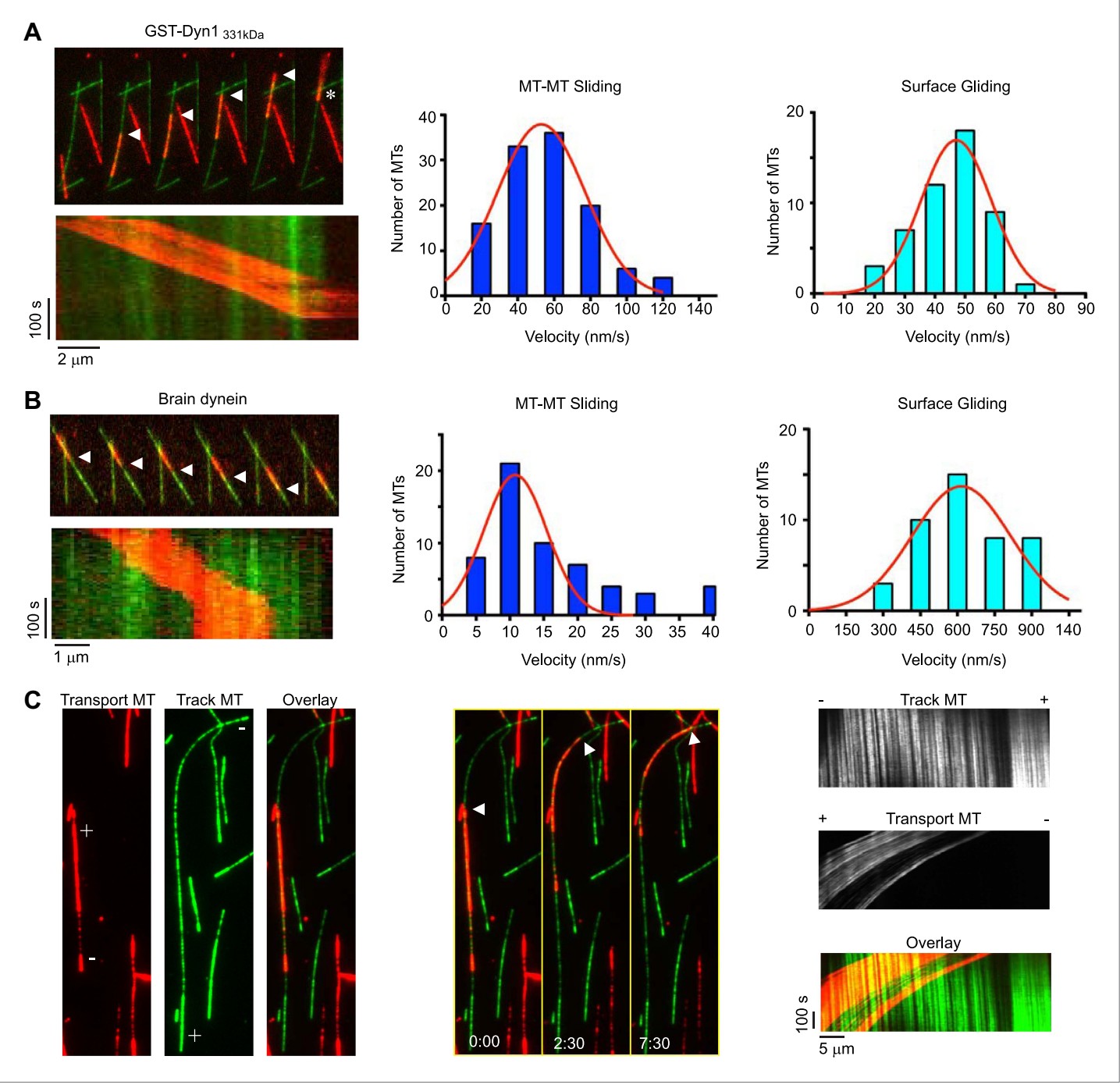

**Figure 2**. Dynein crosslinks and slides single MT overlaps. (**A**) Example of single MT-MT sliding driven by GST-Dyn1₃₃₁kDa. Successive frames, separated by 52 s, from the video and corresponding kymograph show the sliding. The transport MT is captured and aligned onto the track MT by GST-Dyn1₃₃₁kDa. Arrowheads follow the transport MT as it slides along the track MT. Asterisk marks where the transport MT remains attached at the end of the track MT. Right, histograms of the MT-MT sliding and surface gliding velocities driven by GST-Dyn1₃₃₁kDa with Gaussian fitting. (**B**) Example of rat brain dynein driven sliding in a single MT-MT overlap. Successive frames, separated by 46 s from the video are shown with corresponding kymograph below. The transport MT pauses before reaching the end of the track MT, which was frequently observed for rat dynein-driven movement. Velocity histograms for rat dynein driven MT-MT sliding and surface gliding are shown to the right with Gaussian fitting. (**C**) Polarity-marked MTs with long, brightly labeled plus-ends were used to determine the orientation of MT-MT sliding. The plus- and minus-ends of both MTs are indicated. Arrowhead shows the red transport MT slides with its minus-end away from the plus-end of the track MT, indicating that sliding is anti-parallel. Kymograph analysis of the anti-parallel sliding event is shown to the right.

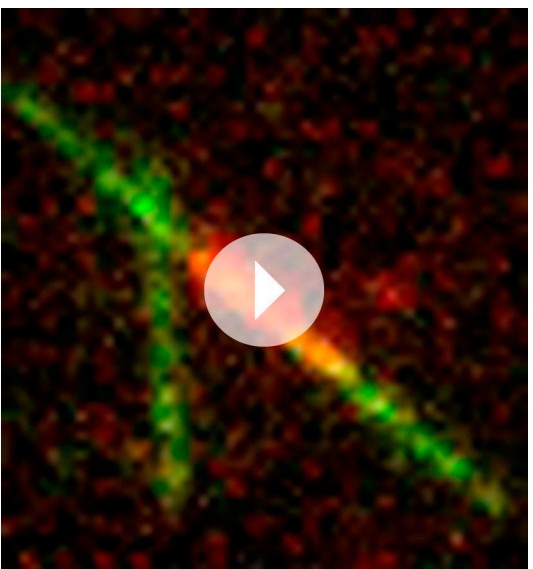

**Video 3**. Rat brain cytoplasmic dynein slides single MT overlaps. A red-labeled 'transport' MT is slid over a green-labeled 'track' MT by rat brain cytoplasmic dynein. The transport MT pauses before reaching the end of the track MT. Total time of the video is 271 s. Playback is 30 fps.

within the spindle (*Blangy et al., 1995*; *Tanenbaum et al., 2009*; *Vanneste et al., 2009*). In the absence of both kinesins, a dynein-dependent inward force causes the rapid collapse of the metaphase spindle to a monopolar structure (*Tanenbaum et al., 2008, 2009*; *Ferenz et al., 2009*; *Vanneste et al., 2009*). While it is still unknown how dynein produces this inward force, it has been speculated to do so by sliding apart anti-parallel MTs (*Tanenbaum et al., 2008*; *Ferenz et al., 2009*). If this is true, then a minimal dynein dimer that elicits anti-parallel MT sliding in vitro might be able to antagonize the outward forces of kinesin-5 and kinesin-12 motors.

To test this hypothesis, we generated a GST-dimerized, GFP-tagged human dynein construct lacking the tail domain (GST-hDyn), similar to yeast GST-Dyn1$_{331kDa}$ construct. GST-hDyn lacks the consensus binding sites for other dynein subunits (*Vallee et al., 2012*), which we confirmed by immunoprecipitation (*Figure 5—figure supplement 1A*). When expressed in HEK293 cells, GST-hDyn partially localized to the spindle (*Figure 5—figure supplement 1B*), indicating that it is a functional MT-binding protein.

To test whether GST-hDyn is able to generate an inward force within the spindle, we designed two types of assays. In the first assay, we evaluated dynein's ability to drive monopolar spindle formation when kinesin-5 is inhibited before the onset of mitosis. Kinesin-5 inhibition results in the formation of monopolar spindles (*Blangy et al., 1995*; *Mayer et al., 1999*), but this effect is prevented and bipolar spindle formation is restored when dynein is depleted by RNAi (*Tanenbaum et al., 2008*; *Ferenz et al., 2009*). This result has been interpreted as kinesin-5 and dynein producing counteracting sliding forces during spindle formation; when kinesin-5 is inhibited, dynein-induced sliding forces produce an unbalanced inward force that results in the formation of monopolar spindes. We tested whether GST-hDyn could produce this inward sliding force when expressed in cells depleted of dynein heavy chain by RNAi. We monitored dynein RNAi by blotting for the intermediate chain (IC), which is co-depleted with the heavy chain upon knockdown and found that dynein was robustly depleted from cells upon RNAi (*Figure 5A*). Strikingly, GST-hDyn expression substantially increased the number of monopolar spindles in dynein-depleted, kinesin-5 inhibited cells (*Figure 5B*). These results reveal that GST-hDyn can recapitulate the inward force, normally produced by the endogenous dynein, providing further support that the motor domains alone are sufficient for this function.

In a second assay, we tested dynein-driven MT sliding forces in metaphase-arrested spindle (cells treated with the proteosome inhibitor MG132). During metaphase, two kinesins (kinesin-5 and kinesin-12) provide an outward force on the spindle, which is counterbalanced by an inward force produced by dynein (*Tanenbaum et al., 2009*; *Vanneste et al., 2009*). Thus, unlike in early stages of mitosis, when a kinesin-5 inhibitor is applied at metaphase, it only results in a collapse of the spindle to a monopolar structure in a low percentage of the cells (38 ± 5%) since kinesin-12 can still effectively oppose dynein (*Figure 5C,D*). However, in metaphase-arrested cells expressing GST-hDyn, kinesin-5 inhibition resulted in a much greater percentage (88 ± 2%) of cells with monopolar spindles (*Figure 5C,D*). While treatment of cells with MG132 might alter the abundance of other MT binding proteins as well in this assay (*Song and Rape, 2010*), this result suggests that the expression of GST-hDyn increased the total inward force, tipping the balance of forces and resulting in spindle collapse. Furthermore, overexpression of GST-hDyn, did not result in spindle collapse in the presence of both kinesin-5 and kinesin-12 activity (*Figure 5—figure supplement 1C*).

Our two assays together suggest that GST-hDyn is sufficient to generate an inward force, which is most likely the product of anti-parallel MT sliding. To determine if dimerization of the motor domains is required for the inward force generation by dynein in vivo, we fused dynein to FKBP (FKBP-hDyn),

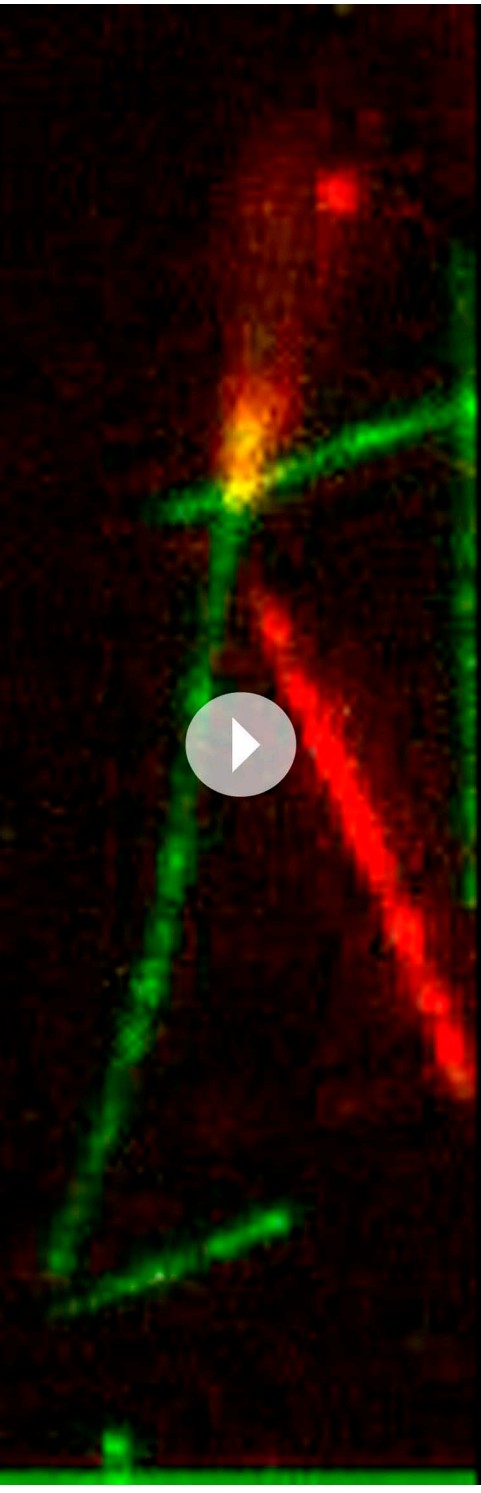

**Video 4**. GST-Dyn1$_{331kDa}$ slides single MT overlaps. A red-labeled 'transport' MT is slid over a green-labeled 'track' MT by GST-Dyn1$_{331kDa}$, indicating that the dynein motor domains alone are sufficient for this activity. The transport MT slides to the end of the track MT and swivels around a nodal attachment point. Total time of the video is 306 s. Playback is 30 fps.

which homodimerizes upon addition of the small molecule dimerizer AP20187 (inducing processivity of yeast dynein [*Reck-Peterson et al., 2006*]), and then performed RNAi rescue experiments. Control cells treated with STLC to inhibit kinesin-5 mostly formed monopolar spindles (87.7 ± 4.5%). As described earlier, depletion of endogenous dynein by RNAi in the presence of STLC decreased the number of monopolar spindles to 46.5 ± 1.5%, consistent with a role for dynein in antagonizing kinesin-5 in bipolar spindle assembly (*Figure 5E*). As a control, the dimerizer AP20187 did not significantly affect spindle bipolarity (50.7 ± 1.8% monopolar spindles, p=0.16) in STLC-treated, dynein-depleted cells in the absence of FKBP-hDyn (*Figure 5E*). Next, cells expressing FKBP-hDyn were depleted of endogenous dynein by RNAi and treated with STLC. Expression of FKBP-hDyn in the absence of AP20187 did not alter the fraction of monopolar spindles (47.8 ± 2.7%; mean ± SD), indicating that the monomeric version of dynein was unable to reconstitute the function of native dynein. However, addition of AP20187 to cells expressing FKBP-hDyn substantially increased in the fraction of cells with monopolar spindles (*Figure 5E*, 75.5 ± 2.5%). This result demonstrates that dimerization of two dynein motor domains robustly activates the ability of dynein to generate an inward force in the spindle. Together, our results suggest that a minimal, dimeric motor can slide anti-parallel MTs to generate an inward force during spindle assembly.

In addition to its role in generating an inward force in the spindle, dynein also is important for focusing MT minus ends at the spindle pole (*Walczak and Heald, 2008*). To test whether the GST-hDyn was able to support spindle pole focusing, we compared spindle pole focusing in control cells, cells depleted of dynein, and cells depleted of dynein that express GST-hDyn. Depletion of dynein increased the percentage of spindles with unfocused poles and detached centrosomes; however, this pole focusing defect was not rescued by GST-hDyn (*Figure 5—figure supplement 1D*). These results show that a minimal dynein is able to generate an inward force in the spindle, which likely involves anti-parallel MT sliding, but is unable to fulfill other mitotic functions of dynein.

## Discussion

### The mechanism of dynein-mediated MT sliding

In this study, we show that cytoplasmic dynein can slide anti-parallel MTs in vitro and provide evidence that this mechanism can occur in vivo. Several kinesin motors have the ability to crosslink and slide MTs in vitro. However, these motors either form a tetrameric complex (*Kashina et al., 1996*; *Kapitein et al., 2008*),

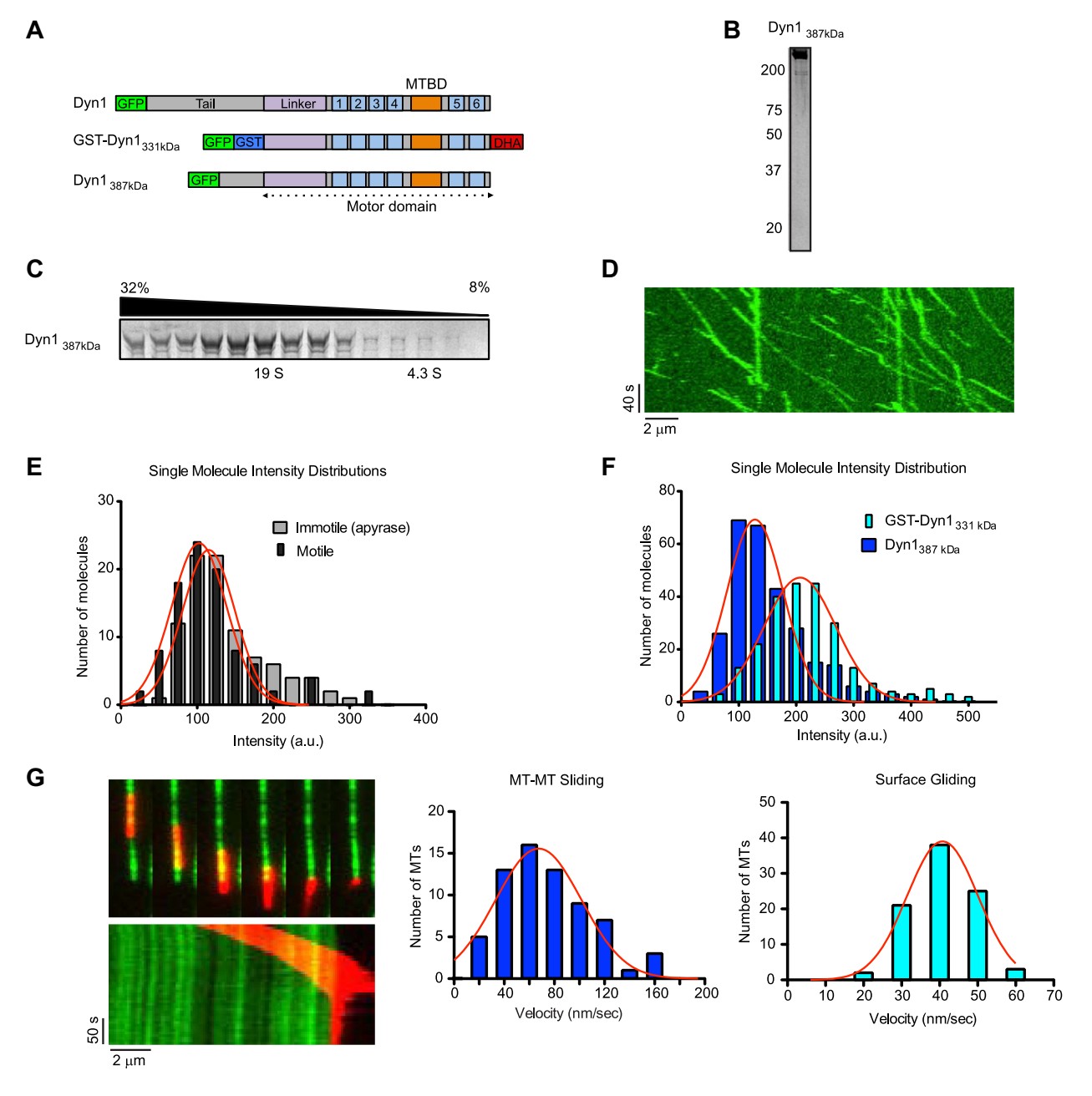

**Figure 3**. Natively dimerized dynein can drive MT-MT sliding in the absence of associated factors. (**A**) Schematic overview of the Dyn1$_{387kD}$ construct compared to Dyn1 and GST-Dyn$_{331kDa}$, with labels as in *Figure 1A*. (**B**) Gel-filtered Dyn1$_{387kD}$ was run on a denaturing gel and proteins were visualized by silver staining. Note the lack of detectable bands at the lower molecular weights expected for dynein accessory subunits. (**C**) Coomassie blue stained gel of sucrose gradient fractions showing sedimentation behavior of Dyn1$_{387kD}$. Position of 19S and 4.3S standards and sucrose concentrations are indicated. (**D**) Kymograph of single Dyn1$_{387kD}$ molecule motility along a MT immobilized on the glass surface. (**E**) Cy5-labeled MTs were surface-immobilized in flow chambers. Dyn1$_{387kD}$ was added either in the presence of ATP or apyrase. Fluorescence intensities of single motors was determined either of all MT-bound molecules in the presence of apyrase (light gray bars), or of the motile motors in the presence of ATP (dark gray bars). (**F**) Flow chambers were prepared as in (**E**), but MTs were incubated with either Dyn1$_{387kD}$ or GST-Dyn1$_{331kD}$ for 5 min in the presence of apyrase. Intensities of GFP spots were quantified after background subtraction. (**G**) Example of Dyn1$_{387kD}$ driven sliding in a single MT-MT overlap. Successive frames, separated by 34 s from the video are shown with corresponding kymograph below. Velocity histograms for MT-MT sliding and surface gliding are shown to the right with Gaussian fitting.

The following figure supplements are available for figure 3:

**Figure supplement 1**. Further characterization of GFP-Dyn1$_{387kD}$.

or contain a second MT binding domain within their tail domain, which facilitates MT crosslinking (*Jolly et al., 2010*) (*Figure 6*). In contrast, our results show that MT crosslinking and sliding by dynein dimers does not require its tail domain, accessory subunits, regulatory proteins, or further oligomerization. Rather, our results indicate that the two motor domains of a dynein dimer can bind to separate MTs and each motor domain can walk independently on the MT to which it is bound. To accomplish this, the two motor domains likely splay apart, allowing the dimer to bind to the two MTs simultaneously (*Figure 6*).

Our single molecule results also reveal that the two dynein motor domains can switch abruptly from walking along the same MT to walking on separate MTs when the motor encounters a MT-MT overlap. Inside the overlap, dynein often pauses and switches directions, suggesting the motor spends a fraction of time stepping with both motor domains on a single MT in the overlap, and some fraction with each motor domain bound simultaneously to opposite MTs. This behavior was observed for both GST-dimerized dynein, as well as two natively dimerized dyneins (Dyn1 and $Dyn1_{387kD}$), indicating that it was not an effect of artificial dimerization. These results suggest that the two motor domains have considerable flexibility, allowing them to explore space on a rapid time scale, making frequent transitions between one and two MT bound states.

The ability of native or artificially dimerized dynein to walk on the same or different MTs suggests that the dynein motor domains work in a more autonomous fashion than kinesin-1. Consistent with this notion, recent data suggests that dynein's two motor domains step in a relatively stochastic and independent fashion (*DeWitt et al., 2012*; *Qiu et al., 2012*). Furthermore, a dynein dimer made of one active and one inactive motor domain, still moves processively on a single MT (*DeWitt et al., 2012*), suggesting that a single active head can step forward, provided that it remains tethered to the MT track. The relatively uncoupled motion of dynein's two motor domains likely allows for them to step independently on oppositely oriented MTs in an overlap and promote MT-MT sliding.

## MT sliding in vivo

Dynein has a key role in minus-end-directed cargo transport along MTs in mammalian cells. In addition, a large body of evidence suggests that dynein controls MT organization during mitosis by sliding MTs within the spindle (*Verde et al., 1991*; *Rusan et al., 2002*; *Tanenbaum et al., 2008*; *Ferenz et al., 2009*; *Gatlin et al., 2009*). In these earlier studies, this MT sliding in vivo was speculated to occur between anti-parallel MTs, generating an inward force that opposes the outward forces of kinesin-5 and kinesin-12 acting upon these overlapping MTs. This model agrees well with our in vitro data showing that dynein virtually always slides MTs in an anti-parallel configuration. Our in vivo experiments also demonstrate that a minimal dimer is able to generate

**Video 5**. $Dyn1_{387kDa}$ slides MTs within bundles. Red-labeled MTs are extruded from a bundle of red- and green-labeled MTs by $Dyn1_{387kDa}$ upon addition of ATP. Total time of the video is 250 s. Playback is 12 fps.

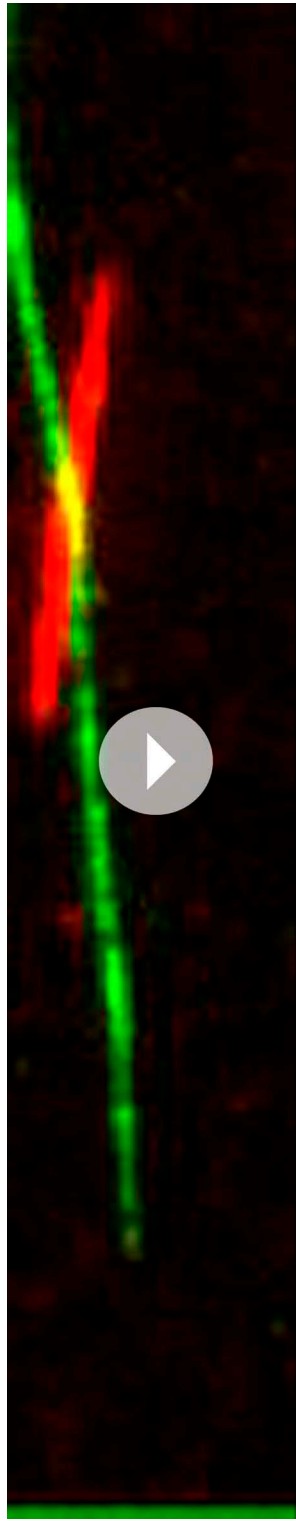

**Video 6**. Dyn1$_{387kDa}$ slides single MT overlaps. A red-labeled 'transport' MT is slid over a green-labeled 'track' MT by Dyn1$_{387kDa}$. Total time of the video is 198 s. Playback is 12 fps.

an inward force in the spindle that can reconstitute the function of the endogenous motor. This activity was observed using two different dimerization methods, indicating that the MT-MT sliding ability is independent of the method of dimerization. As we cannot directly observe MT-MT sliding in the spindle, we cannot exclude that the minimal dynein collapses the spindle through an alternative mechanism, for example through interaction with additional proteins. However, our in vivo analysis is consistent with the notion that dynein slides MTs in the spindle using only its motor domains, which is further supported by the in vitro experiments. These results suggest that sliding of MTs in the spindle reflects an inherent activity of the dynein motor domains alone, consistent with a model in which the two motor domains of dynein walk on two distinct MTs in the spindle, driving their relative movement. However, the situation with native dynein may be more complicated, since other dynein subunits, as well as additional dynein-associated proteins, such as Lis1 and Nde1 contribute to inward force generation in vivo (*Tanenbaum et al., 2008*; *Raaijmakers et al., 2013*). These factors could change the fundamental sliding mechanism of native dynein in vivo, so that the two motor domains do not crossbridge two overlapping microtubules, as described in our model. However, we feel that this is unlikely and rather these factors preserve the fundamental mechanism described (*Figure 6*), but may be necessary for promoting dynein stability, efficient targeting to the spindle, and/or modify dynein's force production (*McKenney et al., 2010*; *Trokter et al., 2012*).

In addition to the sliding mechanism described in this study, dynein may act through multiple parallel mechanisms to control proper MT organization in the spindle. For example, Wühr and Mitchison (*Wühr et al., 2010*) provide evidence that dynein anchored at cytoplasmic sites may also contribute to MT sliding in the spindle in large embryonic cells. Additionally, we show that spindle pole focusing cannot be restored by expression of the minimal dynein dimer. While generation of an inward force likely involves sliding of anti-parallel MTs in the middle of the spindle, pole focusing occurs in a region where most microtubules are expected to have a parallel orientation, perhaps explaining why the minimal dynein is unable to support pole-focusing activity. It is currently unclear how dynein controls pole focusing; however interactions of dynein with NuMA may be important for this activity (*Merdes et al., 1996*, *2000*), Kinesin-14 motors play an important role in pole focusing through MT crossbridging and MT sliding as well (*Figure 6*) (*Mountain et al., 1999*; *Braun et al., 2009*; *Fink et al., 2009*).

MT sliding has also been observed in other cellular systems. While kinesin-1 is best characterized for such activities (*Navone et al., 1992*; *Straube et al., 2006*; *Jolly et al., 2010*), dynein may contribute as well. Anterograde transport of MTs was observed in nerve axons and shown to be dependent upon dynein activity (*He et al., 2005*), although the mechanism by which dynein moved MTs in the axon was unclear. Similarly, loss of dynein in *Drosophila* neurons results in a loss of uniform MT polarity within axons, suggesting that dynein actively sorts MTs based on their polarity (*Zheng et al., 2008*). The mechanism of MT-MT sliding described here provides a model for how dynein may achieve such MT sorting in neuronal cells. Thus, dynein driven MT-MT sliding could be a more generally used mechanism to organize and sort MTs in the cell.

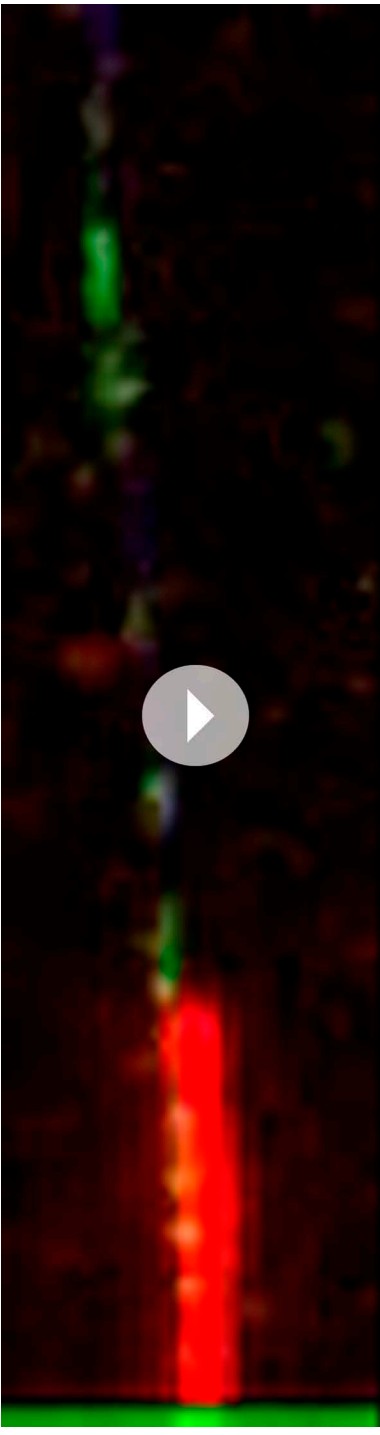

**Video 7**. Observation of single Dyn1$_{387kDa}$ molecules in sliding MT-MT overlaps. Single GFP-labeled Dyn1$_{387kDa}$ molecules visualized during a MT-MT sliding event show homogenous fluorescence, further ruling out protein aggregation as a cause of the sliding behavior. Track MT is blue, transport MT is red. Total time of the video is 120 s. Playback is 6 fps.

## Materials and methods

### Protein preparation

Rat brain cytoplasmic dynein was purified as described (*Paschal et al., 1991*). The peak dynein-containing fractions from the final sucrose gradient were collected, pooled and frozen in LiN2. This preparation of dynein contains no detectable dynactin (*Figure 1A*) (*Ori-McKenney et al., 2010*). Recombinant yeast dynein constructs were purified and labeled with the Halo-TMR ligand as described (*Reck-Peterson et al., 2006*). Dyn1$_{387kD}$ was created by inserting the Gal promoter and ZZ-TEV-HA-GFP cassette (*Reck-Peterson et al., 2006*) into the yeast dynein heavy chain sequence immediately prior to the sequence $_{732}$SYTFYTN. The construct encodes amino acids 732–4092 of the dynein heavy chain. Yeast dynein was gel filtered using a Superose 6 column in gel filtration buffer (50 mM Tris-HCl, pH 8.0, 150 mM K-acetate, 2 mM Mg-acetate, 1 mM EGTA, 10% glycerol, 0.1 mM ATP), except for the full-length dynein which was too low of a concentration and was thus used directly after TEV release.

### Flow chamber preparation

Glass coverslips were acid washed as described (http://labs.bio.unc.edu/Salmon/protocolscoverslippreps.html). A ~10 µl flow chamber was assembled using double-sided sticky tape and a glass slide. The chamber was coated sequentially with the following solutions: 5 mg/ml BSA-biotin (Sigma, St. Louis, MO), 60 µl BC buffer (BRB80, 1 mg/ml BSA, 1 mg/ml casein, 0.5% Pluronic F-168, pH 6.8), 20 µl 0.5 mg/ml streptavidin (Vector Labs, Burlingame, CA), and 60 µl BC buffer to remove excess streptavidin. The chamber was finally washed into assay buffer (30 mM Hepes pH 7.4, 50 mM K-acetate, 2 mM Mg-acetate, 1 mM EGTA, 10% glycerol) containing an oxygen scavenging system (0.5 mg/ml glucose oxidase, 0.1 mg/ml catalase, 25 mM glucose, 70 mM β-mercaptoethanol), 0.2 mg/ml κ-casein, and 0.1% Pluronic F-168.

### MT bundling

Pig brain tubulin was purified and labeled as described (*Castoldi and Popov, 2003*). MTs were assembled using GMP-CPP, centrifuged at 16,000×*g* and resuspended in BRB80 buffer containing 10 µM taxol. MTs were labeled with ~10% fluorescent tubulin and ~10% biotin tubulin where applicable. For bundling experiments, equal volumes of red- or green-labeled MTs were mixed together and then incubated with dynein constructs for 10 min at room temperature. The solution was then perfused into a streptavidin-coated chamber and incubated for 10 min to allow binding to the coverslip surface. The chamber was then washed with BC buffer to remove unbound MTs and imaged using TIRF microscopy. For sliding experiments, assay buffer with 1 mM ATP was added to the chamber and videos were acquired to capture the MT sliding.

### MT sliding

For MT sliding on single overlaps, chambers were first coated with BSA-biotin and streptavidin. Biotinylated track MTs were bound to the surface and the chamber was washed

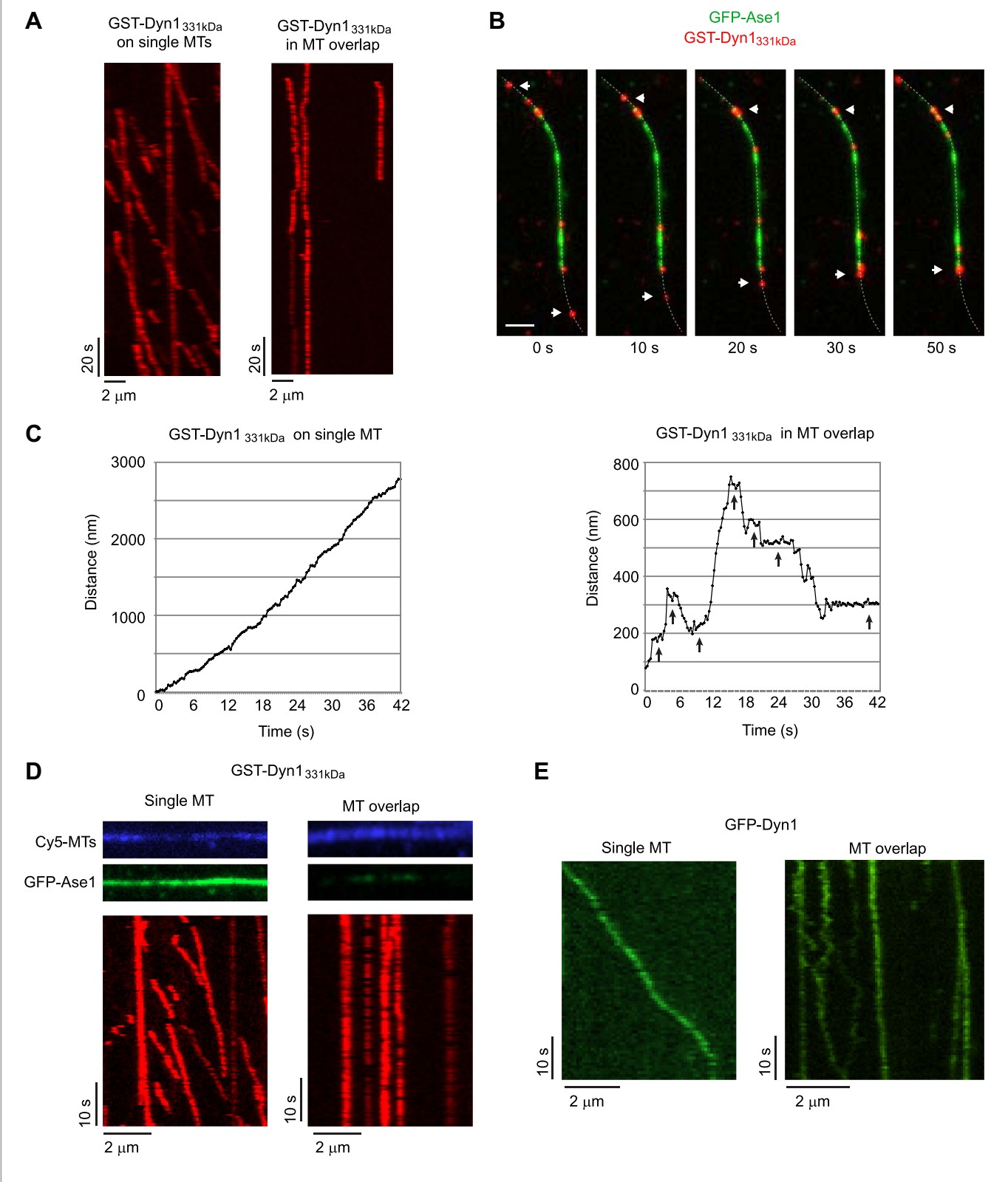

**Figure 4**. Dynein can bind two MTs simultaneously. (**A–E**) Biotinylated Cy5-MTs were incubated with 2 nM GFP-Ase1 or with buffer for 5 min. MTs were then bound to glass slides using surface-bound streptavidin. After washing unbound MTs, TMR-labeled GST-Dyn1$_{331kDa}$ (**A–D**) or GFP-tagged Dyn1 (**E**) was introduced into the flow cell. (**A**) Kymographs of TMR-GST-Dyn1$_{331kDa}$ on single MTs (left panel) and on anti-parallel MT overlaps (right panel).
*Figure 4. Continued on next page*

*Figure 4. Continued*

(**B**) Images from a time series in which individual TMR-GST-Dyn1$_{331kDa}$ molecules can be observed walking into an overlap (visualized by GFP-Ase1) and halting their unidirectional movement. Dotted line indicates the MT. Arrowheads indicate motors that walk from a single MT into an MT overlap. Scale bar is 2 µm. (**C**) High resolution tracking of individual motors on either single MTs (left) or in an anti-parallel MT overlap (right). Arrows indicate long pauses in motility. Tracking precision was ~10 nm. (**D**) Slides were prepared as in (**A**), but 100 nM of GFP-Ase1 was added to single MTs together with TMR-labeled GST-Dyn1$_{331kDa}$. GST-Dyn1$_{331kDa}$ shows unidirectional processive movement on Ase1-coated single MTs, but not in Ase1-generated MT overlaps. (**E**) Slides were prepared as in (**A**), but full-length yeast Dyn1 was used. Kymographs show dynein moving on a single MT (left) or in a MT overlap created by Ase1 (right).

The following figure supplements are available for figure 4:

**Figure supplement 1**. Analysis of single Dyn1$_{387kD}$ molecules in MT overlaps.

extensively to remove unbound MTs. Dynein was then introduced and allowed to bind to the track MTs for 5 min at room temperature. Rat brain dynein at ~100 nM or GST-Dyn1$_{331kDaa}$ at ~200 nM was used. Unbound dynein was removed by washing with BC buffer, followed by introduction of transport MTs diluted in BC buffer. After a 10 min incubation, unbound track MTs were washed out with BC buffer and the chamber was equilibrated in assay buffer containing 2 mM ATP and an oxygen scavenger system. Sliding was imaged in TIRF mode. Data was acquired and subsequently analyzed by making kymographs of the sliding events using µManager. Polarity-marked MTs were made essentially as described previously (*Goodwin and Vale, 2010*), but the GMP-CPP seeds were first capped at their minus ends by incubation with 0.3 µM NEM-treated tubulin. Sliding between MTs was observed in the same buffer with a final concentration of 100 mM K-acetate. At 200 mM K-acetate, no annealing between transport and track MTs was observed.

## High-resolution tracking of single molecules

To track single molecules of dynein with high resolution, the Localization Microscopy plugin of µManager (developed by Nico Stuurman) was used (http://valelab.ucsf.edu/~MM/MMwiki/index.php/Localization_Microscopy).

## Expression of hDyn in HEK293 cells

To express a minimal human dynein constructs in mammalian cells, the pTON vector (a modified version of pcDNA4TO with N-terminal GFP (*Tanenbaum et al., 2011*) was modified to include either N-terminal GFP-Halo-HA-GST tags (GST-hDyn) or GFP-FKBP (F36V) tags (FKBP-hDyn). The C-terminal ~380 kDa motor domain of human dynein (nucleotides 3850-13,938 of clone KIAA1997) was then inserted downstream of either the GST or the FKBP. This boundary was chosen to be similar to the previously published motor domain constructs of rat (*Höök et al., 2005*), Dictyostelium, and yeast dyneins (*Reck-Peterson et al., 2006*). HEK293 cells were transfected with either GST-hDyn or FKBP-hDyn and cells were analyzed ~24 hr after transfection. Expression of FKBP-hDyn at very high expression levels disrupted chromosome alignment and resulted in defects in spindle morphology, independently of AP20187 addition. Very high expressing cells also had an increased tendency to form monopolar spindles. Therefore, for all experiments we focused our analysis on the ~70% of cells within the population with low to moderate expression levels of FKBP-hDyn.

## Sucrose gradient

Step gradients were made by carefully layering 250 µl each of 8, 16, 24, or 32% sucrose in Pipes-Hepes buffer (50 mM Pipes, 50 mM Hepes, 2 mM MgSO$_4$, 1 mM EDTA, pH 7.0) in a TLS-55 tube. A 250 µl solution of either TEV released GFP-Dyn1$_{387kD}$, or gradient standards, was layered on top. The gradients were centrifuged in a TLS-55 rotor at 55K rpm (200,000×*g*) for 3 hr at 4°C. Standards used were thyroglobulin (19S; Sigma) or BSA (4.3S; Sigma). Gradients were fractionated by carefully pipetting 100 µl from the top.

## Quantification of single molecule GFP intensities

Cy5-and biotin-labeled MTs were surface-immobilized using biotin-BSA and streptavidin. GFP-tagged motors were incubated with or without ATP or apyrase in the flow chamber for 5 min, after which images were taken using identical microscopy settings. The density of motors was kept at around 2–4 motors

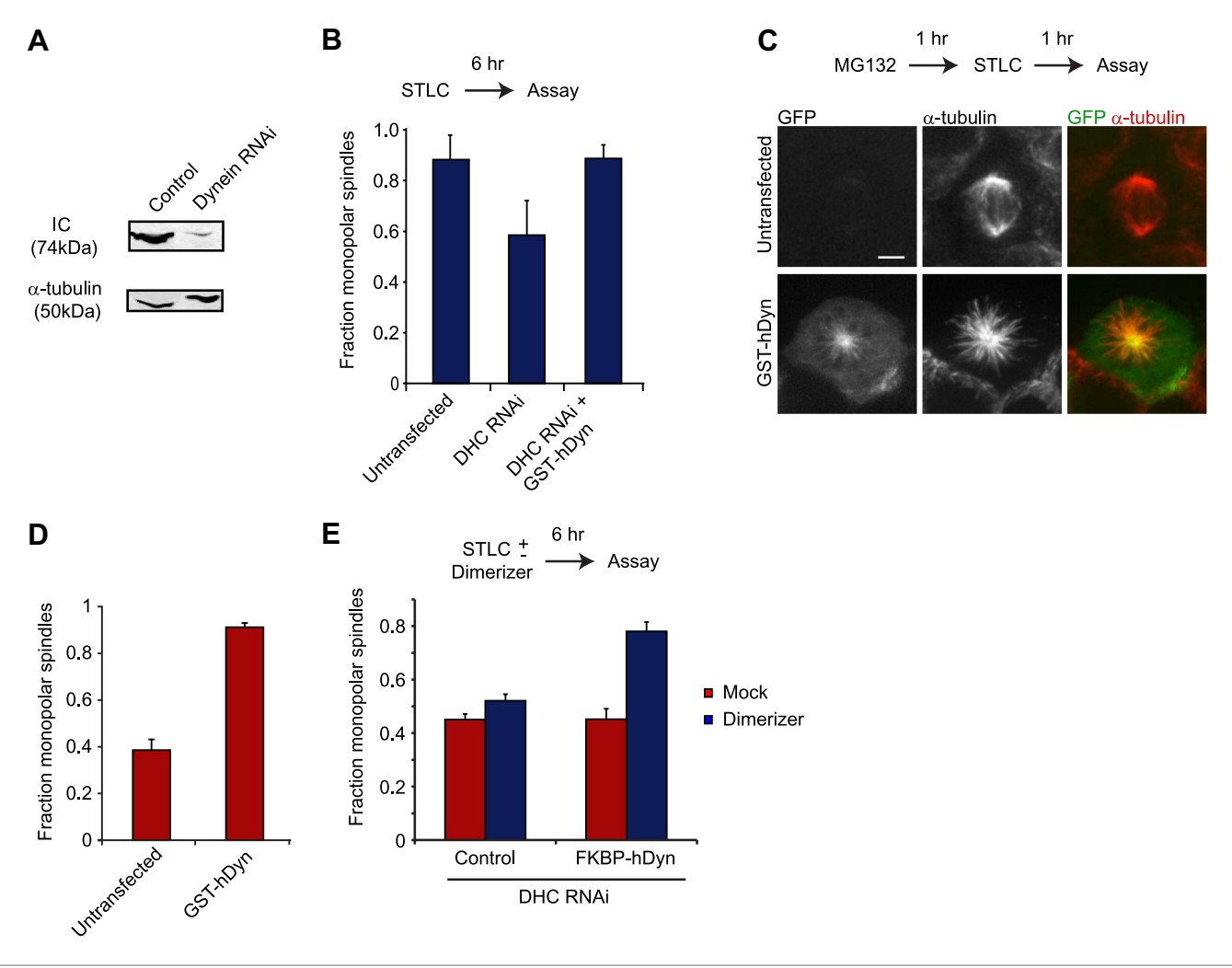

**Figure 5**. Dynein's motor domains are sufficient to produce an inward force in the spindle. (**A**) HEK293 cells were either mock transfected or transfected with siRNA targeting the dynein heavy chain. After 24 hr cells were re-transfected with siRNA. 72 hr after initial transfection, cells were harvested and the level of dynein expression was determined by western blot. Note that the blot was probed for IC, which is co-depleted when dynein heavy chain is depleted. (**B**) Cells were either mock transfected or transfected with siRNA targeting the N-terminus of dynein. After 24 hr, cells were washed and transfected with GFP-tagged GST-hDyn. 48 hr after the first transfection, cells were re-transfected with dynein siRNA. 68 hr after the first transfection, cells were treated with 1 µM STLC for 6 hr and were then fixed and stained for α-tubulin. The fraction of mitotic cells with monopolar spindles was then scored. (**C** and **D**) Cells were transfected with GFP-tagged GST-hDyn. After 24 hr, cells were treated with MG132 for 1 hr and subsequently with 20 µM STLC for 1 hr where indicated. Cells were then fixed and stained for α-tubulin and the percentage of mitotic cells with monopolar spindles was scored. (**C**) shows representative images and (**D**) shows the quantification. (**E**) Cells were treated as in (**C**), except GFP-tagged FKBP-hDyn was transfected instead of GST-hDyn. In addition, 200 nM AP20187 was added together with STLC where indicated. Scale bars, 5 µm. Error bars represent standard deviations. All graphs are averages of three independent experiments with 40–120 cells scored per experiment.

The following figure supplements are available for figure 5:

**Figure supplement 1**. Analysis of GST-hDyn activity in vivo.

per MT to ensure spots represented a single motor. GFP fluorescence was measured in ImageJ and background signal was subtracted.

## Cell culture, transfection, and drug treatments
HEK293 cells were cultured in DMEM supplemented with 10% FCS and antibiotics. siRNA transfections were performed with Hiperfect (Qiagen, Valencia, CA) according to manufacturer's guidelines.

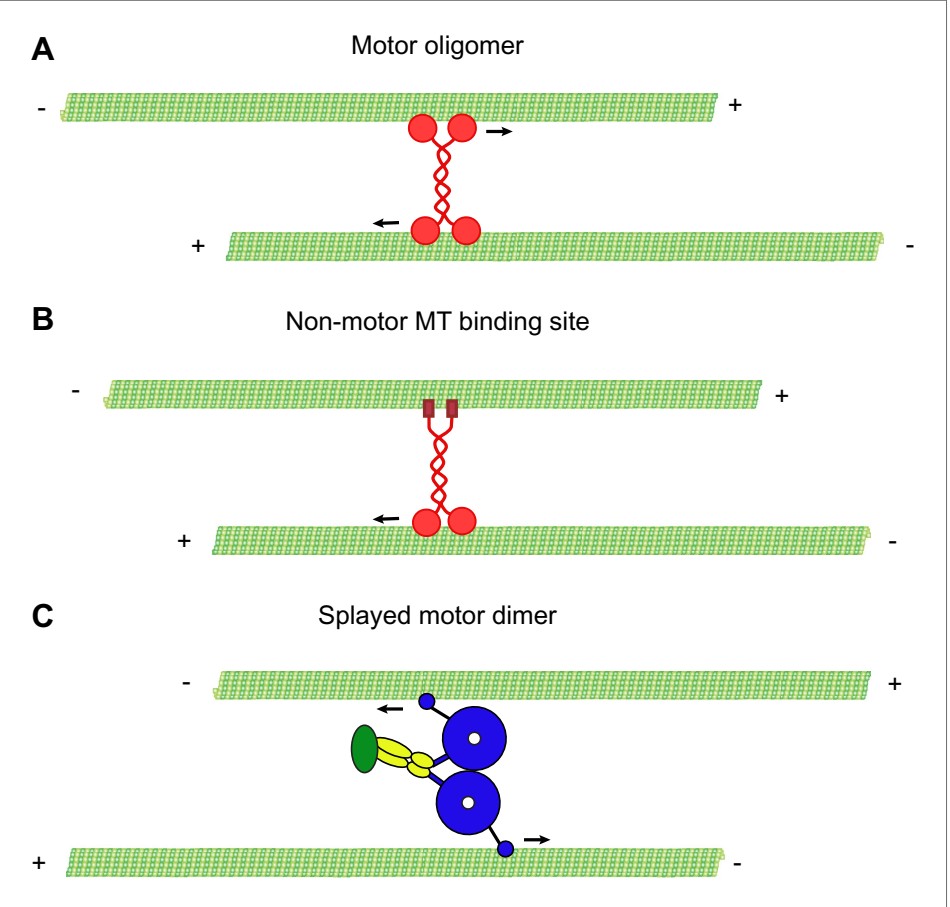

**Figure 6**. Mechanisms of MT-MT sliding. (**A**) Kinesin-5 forms tetrameric molecules, allowing them to crosslink and slide MTs using motor domains at opposite ends of the tetramer. (**B**) Kinesin-1 and kinesin-14 utilize a secondary, non-motor, MT binding site located in their tail domains to crosslink and slide MTs. (**C**) Our data suggest that cytoplasmic dynein utilizes a novel mechanism for crosslinking and sliding MTs. The motor domains of a single dynein dimer splay apart and bind to separate MTs. The motor domains then move on opposite MTs, causing sliding of anti-parallel MTs.

DNA was transfected using polyethyleneimine (PEI). MG132 (Sigma) was dissolved in DMSO and was used at 5 µM final concentration. STLC (Sigma) was dissolved in DMSO and was used at indicated concentrations. AP20187 (Clontech, Mountain View, CA) was dissolved in ethanol and used at a final concentration of 200 nM.

## Immunofluorescence

Cells were grown in glass bottom 96-well plates. At the time of fixation, culture medium was removed and cells were fixed in PBS with 3.7% formaldehyde and 1% Triton X-100 for 5 min. Cells were then washed with PBS and post-fixed with cold methanol for 5 min. Fixed cells were incubated with anti-α-tubulin antibody (1:5000; Sigma) overnight. Secondary antibody was goat-anti-mouse-AlexaFluor555 (1:1000; Invitrogen, Grand Island, NY), which was incubated for 1 hr. Cells were imaged on a Zeiss spinning disc confocal with a 100 × 1.45 NA objective and a Hamamatsu EM-CCD camera. The microscope was controlled by µManager software (*Edelstein et al., 2010*).

## Acknowledgements

We thank Nico Stuurman for help with microscopy and for developing the localization microscopy plugin for µManager. We would also like to thank the Vale lab for helpful discussions.

## Additional information

### Funding

| Funder | Grant reference number | Author |
|---|---|---|
| National Institutes of Health | F32GM096484 | Richard J McKenney |
| Howard Hughes Medical Institute | | Ronald D Vale |
| European Molecular Biology Organization | EMBO ALTF 720-2011 | Marvin E Tanenbaum |
| KWF – Dutch Cancer Society | | Marvin E Tanenbaum |
| National Institutes of Health | R01GM097312 | Ronald D Vale |
| National Institutes of Health | R37GM038499 | Ronald D Vale |

The funders had no role in study design, data collection and interpretation, or the decision to submit the work for publication.

### Author contributions

MET, RJM, Conception and design, Acquisition of data, Analysis and interpretation of data, Drafting or revising the article; RDV, Conception and design, Analysis and interpretation of data, Drafting or revising the article

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
