## [Decision Letter]

[Editors’ note: a previous version of this study was rejected after peer review after concerns that the experiments needed for publication that would represent a substantial amount of work. After completing appropriate revisions, the authors asked for reconsideration. Both decision letters are shown below.]

Thank you for choosing to send your work entitled “Cytoplasmic Dynein Crosslinks and Slides Anti-parallel Microtubules Using Its Two Motor Domains” for consideration at *eLife*. Your article has now been peer reviewed and we regret to inform you that your work will not be considered further for publication. Your submission has been evaluated by 2 reviewers and a member of our Board of Reviewing Editors, and the decision has been discussed further with one of *eLife's* Senior editors.

The Reviewing editor and the outside reviewers discussed their comments before we reached this decision, and the Reviewing editor has assembled the following comments based on the reviewers' reports.

The paper has been seen by three referees. Over all they find the idea that a minimal, GST-dimerized human dynein motor lacking the tail domain and the ability to bind to its adaptor proteins is capable of producing an inward force in the spindle that can substitute for endogenous dynein. These findings are novel. They indeed raise the possibility that dynein-driven MT-MT sliding may be propelled by the two dynein motor domains binding to and walking along two individual MTs.

However, the referees agree that there is one major potential problem in interpreting your data, which comes from artificial dimerization by GST or native rat brain dynein including the three accessory chains and potentially other accessory proteins. The sliding mechanism that the authors present might be an artifact of the artificial dimerization (for example allowing for additional flexibility between the two motor domains), while the native motor could slide the MTs by a different mechanism, for example using a second MT binding site. To show that native dynein slides MTs using the proposed mechanism, the authors could use 'minimal-dimeric-motor' recombinant yeast heavy chains dimerized natively.

We are rejecting the paper because experiments that would be a substantial amount of work should result in a rejection at *eLife*. However, if you feel that you can perform the experiment with the minimal dimeric motor, then we would encourage you to submit your interesting paper again to *eLife*.

A second problem identified by the reviewers concerned whether GST-dynein rescues spindle assembly upon DHC knock-down. This is never shown; rather you show the sensitivity to kinesin-5 inhibition. No data are shown in the absence of the drug, and it is not discussed whether the rescued spindles are positioned correctly and functional to segregate chromosomes, with poles focused, chromosomes aligned, etc. While we would not insist that these data are included, you should more fully discuss the other major functions of dynein in the spindle, and whether they might be accomplished by an anti-parallel sliding mechanism. For example, can this mechanism generate a spindle pole. Again a schematic would help, and ideally some data showing whether or not GST-dynein can generate asters or mediate pole focusing.

*Minor comments*:

- Binding of dynein to MTs without ATP in solution might impair further functionality of the motors. The authors could address this by performing a single molecule no-ATP to ATP switching experiment on single MTs and quantify the percentage of processively moving dynein molecules.

- The native dynein slides MTs very slowly and with pauses. Are the pauses accounted for in the average velocity in the histogram in Figure 2? What is the explanation of the slow sliding speed compared to the fast speed of the surface gliding?

- The fact that the GST-hDyn slides MTs in vivo doesn't clarify the mechanism of the native dynein sliding.

- Using polarity-marked MTs an anti-parallel MT orientation was inferred for 18 out of 21 events. Were the other 3 events parallel sliding? How good was the polarity marking? Please quantify.

- What influence did the ionic strength of the assay buffer have on the results?

- The role of kinesin-14 as minus-end directed, MT-MT sliding motor in spindle assembly and integrity should be discussed.

- In the Introduction, it is stated that there is no evidence that dynein could directly cross link and slide MTs relative to one another. This may be true for pure motor domains in vitro, but movements of MT seeds on the spindle in *Xenopus* egg extracts, and spindle pole focusing, have been shown to require dynein. Similarly, “These results provide the first direct demonstration that cytoplasmic dynein can slide MTs within bundles” seems to be a bit of an overstatement.

- One issue that speaks to whether dynein could carry microtubules as cargo is the presence of non-motor microtubule binding domains. While perhaps not present on the dynein motor itself, doesn't the fact that rat dynein possesses a much slower velocity on bundled MTs compared to single microtubules indicate there is another MT binding domain on the heavy chain? Furthermore, spindle dynein is known to function together with dynactin, which does have a MT binding domain. The authors show convincingly that just a motor dimer can mediate one aspect of spindle morphogenesis, the inward sliding, but without examining the other functions they should soften their statements about the sufficiency of this mechanism.

[Editors’ note: what now follows is the decision letter after additional work had been performed.]

Thank you for giving us another opportunity to evaluate your work entitled “Cytoplasmic dynein crosslinks and slides anti-parallel microtubules using its two motor domains” for consideration at *eLife*. Your article has been favorably evaluated by a Senior editor, a Reviewing editor, and 2 reviewers, one of whom, Stefan Diez, wants to reveal his identity.

The Reviewing editor and the two reviewers discussed their comments before we reached this decision, and the Reviewing editor has assembled the following comments to help you prepare a revised submission.

Although the results are much stronger [than those in the previous submission], we still have some concerns about the experiments that we would like you to address before publication. These address both the in vitro and the in vivo experiments.

*On the side of the cell work*:

1) The authors switch to using GST-GFP-hDyn for the cell experiments but do not describe this construct very well beyond which nucleotides were sub cloned in the Materials and methods section. How similar/identical is it to the yeast protein? Why can it be assumed to behave the same? Does it associate with any other proteins within cells? How does it run on a sucrose gradient/gel filtration? Does the yeast protein behave similarly when introduced? Does the dimerized yeast construct rescue a yeast dynein mutant?

2) To what degree is endogenous dynein depleted? Blots should be shown also to show how much overexpression there is of the introduced proteins.

3) The two assays in Figure 5 are basically showing the same thing twice. The authors should note that MG132 treatment does more than just arrest at metaphase as it influences turnover of MAPs (Song and Rape, Mol. Cell 2011).

4) In Figure 5, the fraction of monopolar spindles in response to STLC in an RNAi control should be shown.

5) In Figure 5—figure supplement 1, the spindles appear more multipolar than with “split poles” and the chromosomes are misaligned. How is it determined whether the cells are in metaphase or prometaphase and does dynein RNAi cause chromosome misalignment?

*On the in vitro work*:

1) In order to exclude dynein aggregation as a potential cause for MT bundling and sliding, the authors should still investigate and show the dynein-GFP signal during the MT–MT sliding experiments (Figure 1, and Figure 2). Does the signal look homogeneous ruling out the possibility of clustering? The authors probably already have the data to answer this question.

To exclude the protein aggregation, the authors now did look already at single GFP labelled dynein molecules (new Figure 3). However, the dynein concentration in single molecule experiments is most likely much lower than in sliding experiments, so the lack of aggregates in single molecule experiments still doesn’t rule out that there are clusters in the sliding assay.

---

## [Author Response]

*[Editors’ note: the author responses to the first round of peer review follow.*]

*The paper has been seen by three referees. Over all they find the idea that a minimal, GST-dimerized human dynein motor lacking the tail domain and the ability to bind to its adaptor proteins is capable of producing an inward force in the spindle that can substitute for endogenous dynein. These findings are novel. They indeed raise the possibility that dynein-driven MT-MT sliding may be propelled by the two dynein motor domains binding to and walking along two individual MTs*.

*However, the referees agree that there is one major potential problem in interpreting your data, which comes from artificial dimerization by GST or native rat brain dynein including the three accessory chains and potentially other accessory proteins. The sliding mechanism that the authors present might be an artifact of the artificial dimerization (for example allowing for additional flexibility between the two motor domains), while the native motor could slide the MTs by a different mechanism, for example using a second MT binding site. To show that native dynein slides MTs using the proposed mechanism, the authors could use 'minimal-dimeric-motor' recombinant yeast heavy chains dimerized natively*.

*We are rejecting the paper because experiments that would be a substantial amount of work should result in a rejection at eLife. However, if you feel that you can perform the experiment with the minimal dimeric motor, then we would encourage you to submit your interesting paper again to eLife*.

We have now addressed this critical point in full in two new figures (Figure 3 and Figure 3—figure supplement 1). As suggested by the reviewers, we expressed and purified a natively dimerized recombinant minimal dynein (Dyn_387kD_). We show that Dyn_387kD_ sediments at ∼19S on a sucrose gradient, similar to other dimeric dyneins, and does not contain additional subunits. By deleting its canonical MT binding domain, we also find no evidence for a second “cryptic” MT binding site that might crosslink microtubules. Functionally, single Dyn_387kD_ molecules move processively along MTs. Next, we show that Dyn_387kD_ can bundle MTs, slide MTs within the bundles, and slide MTs within single MT-MT overlaps, very similar to GST-Dyn_331kD_. While Dyn_387kD_ walks processively and unidirectionally on single MTs, it frequent pauses and reverses direction in anti-parallel MT overlaps. Taken together, these results show that a minimal, natively dimerized motor can slide MTs using only its two motor domains. We should also note that transition from unidirectional motion along a single microtubule, to pauses and bidirectional runs within an overlap is most likely a single molecule signature of the two motors domains binding to two different microtubules in an overlap region. This behavior is noted for native yeast dynein, again underscoring that this behavior is not an artifact of truncating the heavy chain.

*A second problem identified by the reviewers concerned whether GST-dynein rescues spindle assembly upon DHC knock-down. This is never shown; rather you show the sensitivity to kinesin-5 inhibition. No data are shown in the absence of the drug, and it is not discussed whether the rescued spindles are positioned correctly and functional to segregate chromosomes, with poles focused, chromosomes aligned, etc. While we would not insist that these data are included, you should more fully discuss the other major functions of dynein in the spindle, and whether they might be accomplished by an anti-parallel sliding mechanism. For example, can this mechanism generate a spindle pole. Again a schematic would help, and ideally some data showing whether or not GST-dynein can generate asters or mediate pole focusing*.

We thank the referees for this clarification, as we did not describe our results and views adequately. We never meant to imply that a minimal dynein dimer can fully compensate for all dynein functions in spindle assembly. Our results show that GST-dynein (or chemically dimerized FKBP-FRB-dynein) can counteract the outward microtubule sliding forces generated by kinesin-5 and kinesin-12, which strongly suggests that this construct produces anti-parallel microtubule sliding in vivo. However, we agree that other dynein functions in mitosis are unlikely to be replicated by this minimal dynein dimer, including pole focusing which occurs mainly between parallel microtubules. We have now tested this directly and show that GST-dynein does not rescue spindle pole focusing after dynein heavy chain RNAi. The spindle pole focusing mechanism likely involves the dynein tail domain, as well as associated molecules such as NuMa. We have clarified these points in the Discussion. We also have expanded the Discussion with recent data showing that the inward sliding forces produced by native dynein in the spindle require associated chains. While we feel that native dynein and the minimal GST-dynein slide anti-parallel MTs by the same mechanism, these results illustrate that native dynein has additional requirements for in vivo function, which may include stability of the entire heavy chain by associated proteins, as well as regulation of its force-generating activities. Thus, between our new in vivo data and new Discussion section, we feel that we have better represented the broader activities of dynein in the spindle. We experimented with a schematic but felt that it did not enhance the paper. We have added a schematic of our model for dynein-driven sliding and how it differs from mechanisms proposed for other motor proteins.

*Minor comments*:

*- Binding of dynein to MTs without ATP in solution might impair further functionality of the motors. The authors could address this by performing a single molecule no-ATP to ATP switching experiment on single MTs and quantify the percentage of processively moving dynein molecules*.

While this is an interesting suggestion, we are unfortunately unable to do such experiments because we do not have a setup to add ATP to our flow chamber while performing simultaneous imaging. We note that the majority of dynein molecules appear to be motile in our assays.

*- The native dynein slides MTs very slowly and with pauses. Are the pauses accounted for in the average velocity in the histogram in*
Figure 2*? What is the explanation of the slow sliding speed compared to the fast speed of the surface gliding*?

MTs that were stationary for prolonged periods of time were not included in the average speed. However, since our time between images in these experiments was 3-5 sec, pauses of less than 3 sec are likely included in the average speed. Nonetheless, these pauses cannot explain the slow speed of MT-MT sliding. We currently do not fully understand why rat brain dynein slides MTs faster in surface gliding experiments than in MT-MT sliding. One possible explanation is that mammalian dyneins have a low MT affinity, much lower processivity, and show birectional movements in vitro (Trokter et al., 2012; Ori-McKenney et al. 2010, and our own unpublished results), while yeast dynein is more processive and unidirectional. This point is now discussed in the manuscript.

*- The fact that the GST-hDyn slides MTs in vivo doesn't clarify the mechanism of the native dynein sliding*.

Our findings that native rat bovine brain dynein and four different recombinant dimerized dyneins (two yeast and two human) all can slide MTs make it reasonable to propose a model in which native dynein slides MTs by a similar mechanism. However, we cannot exclude the possibility that native dynein uses a sliding mechanism that differs from what we have observed in our in vitro assay. We have noted this possibility in the Discussion.

*- Using polarity-marked MTs an anti-parallel MT orientation was inferred for 18 out of 21 events. Were the other 3 events parallel sliding? How good was the polarity marking? Please quantify*.

Polarity marked MT can be problematic, since occasionally they fragment or anneal to yield a spurious result. In this revision, we have included an additional method of assessing polarity- direct single molecule observation of the direction of movement of individual GFP-dynein molecules on the track and transport MTs. Using this approach we found 21/22 sliding events to be anti-parallel. The one parallel event showed very slow sliding. This analysis is now included in the manuscript. Collectively, these two methods reveal predominant anti-parallel MT sliding with an occasional MT sliding event.

*- What influence did the ionic strength of the assay buffer have on the results*?

We tested MT-MT sliding at a higher ionic strength (100 mM KAc) and observed very similar sliding activity. At 200mM KAc, we observed no crosslinking or sliding of MTs. We included this information in the Materials and methods section.

*- The role of kinesin-14 as minus-end directed, MT-MT sliding motor in spindle assembly and integrity should be discussed*.

We added a sentence indicating kinesin-14’s important role in pole focusing to the Discussion.

*- In the Introduction, it is stated that there is no evidence that dynein could directly cross link and slide MTs relative to one another. This may be true for pure motor domains in vitro, but movements of MT seeds on the spindle in Xenopus egg extracts, and spindle pole focusing, have been shown to require dynein. Similarly, “These results provide the first direct demonstration that cytoplasmic dynein can slide MTs within bundles” seems to be a bit of an overstatement*.

It is true that previous studies found that dynein is required for MT movement and organization in the spindle (which we stated in the text, including references). However, it has never been shown that dynein alone is sufficient for sliding of MTs in the spindle, as it could be indirect and require additional factors. Nonetheless, we have changed the text to specify that our results are the first fully reconstituted in vitro demonstration of dynein-dependent sliding and have added a reference to the study that shows transport of stabilized seeds to the spindle pole by dynein (Heald et al., 1996).

*- One issue that speaks to whether dynein could carry microtubules as cargo is the presence of non-motor microtubule binding domains. While perhaps not present on the dynein motor itself, doesn't the fact that rat dynein possesses a much slower velocity on bundled MTs compared to single microtubules indicate there is another MT binding domain on the heavy chain*?

We have now included new data, as described above, showing that a minimal, natively dimerized yeast dynein molecule can drive MT-MT sliding in the absence of associating factors and without an additional MT binding site. We do not fully understand why rat dynein possesses a slower MT-MT sliding speed. It is possible that this is due to an additional MT-binding site somewhere in the complex (which may be regulated in vivo), but it is also possible that this is due to the short dwell time and low processivity of mammalian dynein in vitro (Trokter et al., 2012 PNAS, Miura et al. 2010 FEBS Letters, our unpublished results) and we have added these possible reasons to the Results section.

*- Furthermore, spindle dynein is known to function together with dynactin, which does have a MT binding domain*.

As noted above, we cannot completely exclude that the native dynein complex has an extra MT binding site, but MT-MT sliding in the spindle appears to be independent of dynactin (Raaijmakers et al., 2013, JCB).

*- The authors show convincingly that just a motor dimer can mediate one aspect of spindle morphogenesis, the inward sliding, but without examining the other functions they should soften their statements about the sufficiency of this mechanism*.

As discussed above, we have now included additional data examining the ability of the minimal dimer to rescue spindle pole focusing. We have adjusted our statements in the abstract and main text regarding the role of dynein-dependent MT sliding in spindle morphogenesis accordingly.

*[Editors’ note: the author responses to the re-review follow.*]

*On the side of the cell work*:

*1) The authors switch to using GST-GFP-hDyn for the cell experiments but do not describe this construct very well beyond which nucleotides were sub cloned in the Methods section. How similar/identical is it to the yeast protein? Why can it be assumed to behave the same? Does it associate with any other proteins within cells? How does it run on a sucrose gradient/gel filtration? Does the yeast protein behave similarly when introduced? Does the dimerized yeast construct rescue a yeast dynein mutant*?

We now provide more detailed information on this construct in the Materials and methods. The GST-GFP-hDyn construct is based on previously published boundaries for the minimal motor domains in yeast, *Dictyostelium*, and rat ([49], Koonce and Samso, 1996, and [23]). The human and yeast constructs used in this study are truncated at similar points relative to the motor domain (human residue number 1284 and yeast residue number 1219) and fused to GST. This truncation lies 189 and 145 residues N-terminal to the beginning of linker subdomain 1 in human and yeast respectively. This construct lacks the tail domain of dynein, and thus is not expected to associate with any other dynein subunits. To demonstrate this, we have now included a pull-down experiment demonstrating that the GST-GFP-hDyn construct indeed does not associate with other dynein subunits (Figure 5—figure supplement 1). In other ongoing work in our lab, we have expressed a similar human GST-dynein construct using the baculovirus system and show by hydrodynamic studies (sucrose gradient/gel filtration) and negative stain EM that it is indeed a dimer, as expected based upon its fusion to a GST dimer. We believe the other suggestions, while interesting, are beyond of the scope of our study. Also, the truncated GST-dynein dimer will not have the associated subunits needed to rescue dynein function in spindle positioning in yeast (Markus et al. 2009).

*2) To what degree is endogenous dynein depleted? Blots should be shown also to show how much overexpression there is of the introduced proteins*.

We have now included a western blot showing that dynein is strongly down regulated by ∼87% upon RNAi (Figure 5). This result agrees with prior work on dynein reduction by RNAi (60). Unfortunately, we are unable to determine the precise level of overexpression because the minimal dimer lacks the N-terminal domain, which is targeted by all commercially available antibodies to dynein heavy chain. Furthermore, because the transfection efficiency in our overexpression experiments is only 5–10%, and varies between cells, blotting cell lysates from the total population would not be very informative on the amount of overexpression per cell. We state in the Materials and methods section that we focused on cells with moderate expression levels and exclude the cells with very high expression.

*3) The two assays in*
Figure 5
*are basically showing the same thing twice. The authors should note that MG132 treatment does more than just arrest at metaphase as it influences turnover of MAPs (Song and Rape, Mol. Cell 2011)*.

We have now included the reference to the turnover of MAPs in response to MG132 treatment. We included two different assays, exactly for this reason. As each assays has its specific pitfalls, we performed two assays to be sure that the ability of dynein to collapse the spindle was not due to a specific type of assay.

*4) In*
Figure 5*, the fraction of monopolar spindles in response to STLC in an RNAi control should be shown*.

This information is presented in Figure 5.

*5) In*
Figure 5—figure supplement 1*, the spindles appear more multipolar than with “split poles” and the chromosomes are misaligned. How is it determined whether the cells are in metaphase or prometaphase and does dynein RNAi cause chromosome misalignment*?

The multipolar spindles are due to centrosomes that detach from the spindle, which is a well-known phenotype of dynein depletion. Similarly, dynein depletion is known to cause chromosome alignment defects (Draviam et al., 2006; [47]). Cells in metaphase and prometaphase were distinguished based on chromosome alignment to the metaphase plate, as visualized by DNA staining.

*On the in vitro work*:

*1) In order to exclude dynein aggregation as a potential cause for MT bundling and siding, the authors should still investigate and show the dynein-GFP signal during the MT-MT sliding experiments (*Figure 1
*and*
Figure 2*). Does the signal look homogeneous ruling out the possibility of clustering? The authors probably already have the data to answer this question*.

*To exclude the protein aggregation, the authors now did look already at single GFP labeled dynein molecules (new*
Figure 3*). However, the dynein concentration in single molecule experiments is most likely much lower than in sliding experiments, so the lack of aggregates in single molecule experiments still doesn't rule out that there are clusters in the sliding assay*.

We have included a representative Video 7 to show the GFP-dynein signal during a MT sliding event. The dynein signal is homogenous and does not show any evidence of aggregation.